# Online POMDP Planning with Anytime Deterministic Guarantees

**Moran Barenboim**
Technion Autonomous Systems Program (TASP)
Technion - Israel Institute of Technology
NVIDIA
`moranbar@campus.technion.ac.il`

**Vadim Indelman**
Department of Aerospace Engineering
Technion - Israel Institute of Technology
`vadim.indelman@technion.ac.il`

## Abstract

Autonomous agents operating in real-world scenarios frequently encounter uncertainty and make decisions based on incomplete information. Planning under uncertainty can be mathematically formalized using partially observable Markov decision processes (POMDPs). However, finding an optimal plan for POMDPs can be computationally expensive and is feasible only for small tasks. In recent years, approximate algorithms, such as tree search and sample-based methodologies, have emerged as state-of-the-art POMDP solvers for larger problems. Despite their effectiveness, these algorithms offer only probabilistic and often asymptotic guarantees toward the optimal solution due to their dependence on sampling. To address these limitations, we derive a deterministic relationship between a simplified solution that is easier to obtain and the theoretically optimal one. First, we derive bounds for selecting a subset of the observations to branch from while computing a complete belief at each posterior node. Then, since a complete belief update may be computationally demanding, we extend the bounds to support reduction of both the state and the observation spaces. We demonstrate how our guarantees can be integrated with existing state-of-the-art solvers that sample a subset of states and observations. As a result, the returned solution holds deterministic bounds relative to the optimal policy. Lastly, we substantiate our findings with supporting experimental results.

## 1 Introduction

Partially Observable Markov Decision Processes (POMDPs) serve as a comprehensive mathematical framework for addressing uncertain sequential decision-making problems. Despite their applicability, most problems framed as POMDPs struggle to achieve optimal solutions, largely due to factors such as large state spaces and an extensive range of potential future scenarios. The latter tends to grow exponentially with the horizon, rendering the solution process computationally prohibitive. The advent of approximate online, tree-based solvers has expanded the capacity of POMDPs, enabling them to tackle larger problems by providing a more scalable approach to problem-solving.

A prominent search algorithm addressing the challenges posed by large state and observation spaces in POMDPs is POMCP [Silver and Veness, 2010]. POMCP is a forward search algorithm which handles the large state and observation spaces by aggregating Monte-Carlo rollouts of future scenarios in a tree structure. During each rollout, a single state particle is recursively propagated from the root node to the leaves of the tree. It adaptively trades off between actions that lead to unexplored areas of the tree and actions that lead to rewarding areas of the tree search by utilizing UCT [Auer et al., 2002]. The guarantees on the provided solution by POMCP are asymptotic, implying that the quality of the solution remains unknown within any finite time frame.

37th Conference on Neural Information Processing Systems (NeurIPS 2023).

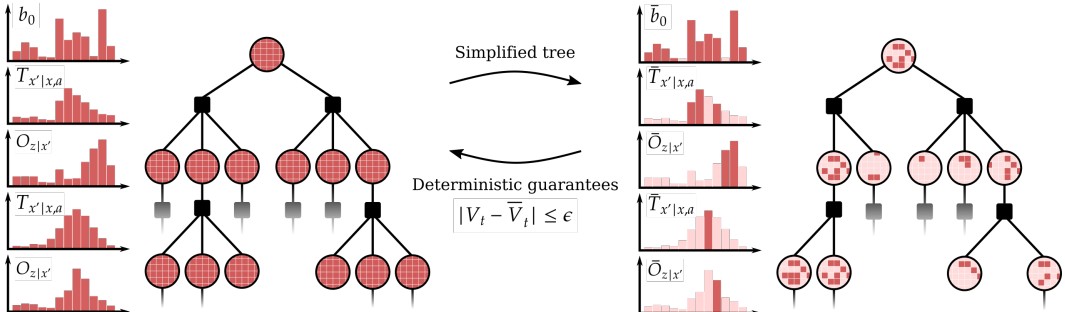

Figure 1: The figure depicts two search trees: a complete tree (left) that considers all states and observations at each planning step, and a simplified tree (right) that incorporates only a subset of states and observations, linked to simplified models. Our methodology establishes a deterministic link between these two trees.

Another notable approximate solver, Anytime Regularized DESPOT (AR-DESPOT) [Somani et al., 2013, Ye et al., 2017] is derived from Regularized DESPOT, which holds theoretical guarantees for the solution quality with respect to its optimal value. Similar to POMCP, AR-DESPOT performs forward search and propagates a single particle from the root node down to its leaves. It relies on branch-and-bound approach in the forward search, and utilizes dynamic programming techniques to update the value function estimate at each node. In contrast to POMCP, Regularized DESPOT offers a probabilistic lower bound on the value function obtained at the root node, providing a theoretical appeal by measuring its proximity to the optimal policy.

While the primary focus of this paper is on discrete POMDP planning, it is essential to acknowledge recent advancements in POMDP planning that encompass both discrete and continuous observation spaces. Few notable approaches include POMCPOW [Sunberg and Kochenderfer, 2018], LABECOP [Hoerger and Kurniawati, 2021] and AdaOPS [Wu et al., 2021], which leverage explicit use of observation models. These algorithms employ importance sampling mechanisms to weigh each state sample based on its likelihood value, which is assumed to be known. Although these methods have exhibited promising performance in practical scenarios, they currently lack formal guarantees. To address this gap, [Lim et al., 2020, 2022] introduced a simplified solver aimed at bridging the theoretical gap between the empirical success of these algorithms and the absence of theoretical guarantees for continuous observation spaces. In [Lim et al., 2022], probabilistic guarantees were derived for the simplified solver concerning its proximity to the optimal value function, thus contributing to a more comprehensive understanding of POMDP planning in both discrete and continuous settings.

In this paper, we focus on deriving deterministic guarantees for POMDPs with discrete state and observation spaces. Unlike existing black-box sampling mechanisms employed in algorithms such as [Sunberg and Kochenderfer, 2018, Hoerger and Kurniawati, 2021, Wu et al., 2021], our approach assumes access not only to the observation model but also to the transition model and the prior. By leveraging this additional information, we develop a novel algorithm that utilizes a subset of the observation space, enabling the computation of deterministic bounds with respect to the optimal policy at any belief node within the constructed tree. Furthermore, we demonstrate how these deterministic bounds can be integrated with state-of-the-art algorithms, including those that sample subsets of both the state and observation spaces. In particular, we provide experimental results showcasing the use of our bounds incorporated into the AR-DESPOT algorithm.

In this paper, our main contributions are as follows. First, we derive deterministic upper and lower bounds for a POMDP problem by considering a subset of the observation space at each node along the tree under a fixed policy. Next, we extend these bounds to cover scenarios where both the state and observation spaces are restricted to subsets, enhancing the applicability of our bounds in practical settings. Based on the derived bounds, we illustrate how to incorporate the bounds into a general structure of common state-of-the-art algorithms. Specifically, in the experimental section, we present results demonstrating the integration of our theoretical guarantees with the AR-DESPOT algorithm, yielding certified solutions with deterministic bounds. Last, we perform simulations to demonstrate the use of our derived bounds in practice, further validating their relevance in POMDP planning.

## 2 Preliminaries

A finite horizon POMDP $M$ is defined as a tuple $\langle \mathcal{X}, \mathcal{A}, \mathcal{Z}, T, O, \mathcal{R} \rangle$, where $\mathcal{X}$, $\mathcal{A}$, and $\mathcal{Z}$ represent a discrete state, action, and observation spaces, respectively. The transition density function $T(x_t, a_t, x_{t+1}) \triangleq \mathbb{P}(x_{t+1}|x_t, a_t)$ defines the probability of transitioning from state $x_t \in \mathcal{X}$ to state $x_{t+1} \in \mathcal{X}$ by taking action $a_t \in \mathcal{A}$. The observation density function $O(x_t, z_t) \triangleq \mathbb{P}(z_t|x_t)$ expresses the probability of receiving observation $z_t \in \mathcal{Z}$ from state $x_t \in \mathcal{X}$.

Given the limited information provided by observations, the true state of the agent is uncertain and a probability distribution function over the state space, also known as a belief, is maintained. The belief depends on the entire history of actions and observations, and is denoted $H_t \triangleq \{z_{1:t}, a_{0:t-1}\}$. We also define the propagated history as $H_t^- \triangleq \{z_{1:t-1}, a_{0:t-1}\}$. At each time step $t$, the belief is updated by applying Bayes' rule using the transition and observation models, given the previous action $a_{t-1}$ and the current observation $z_t$, $b(x_t) = \eta_t \mathbb{P}(z_t|x_t) \sum_{x_{t-1} \in \mathcal{X}} \mathbb{P}(x_t|x_{t-1}, a_{t-1}) b(x_{t-1})$, where $\eta_t$ denotes a normalization constant and $b_t \triangleq \mathbb{P}(x_t \mid H_t)$ denotes the belief at time t. The updated belief, $b_t$, is sometimes referred to as the posterior belief, or simply the posterior. We will use these interchangeably throughout the paper.

A policy function $a_t = \pi_t(b_t)$ determines the action to be taken at time step $t$, based on the current belief $b_t$ and time $t$. In the rest of the paper we write $\pi_t \equiv \pi_t(b_t)$ for conciseness. The reward is defined as an expectation over a state-dependent function, $r(b_t, a_t) = \mathbb{E}_{x \sim b_t}[r_x(x, a_t)]$, and is bounded by $-\mathcal{R}_{\max} \le r_x(x, a_t) \le \mathcal{R}_{\max}$. The value function for a policy $\pi$ over a finite horizon $\mathcal{T}$ is defined as the expected cumulative reward received by executing $\pi$ and can be computed using the Bellman update equation,

$$V_t^\pi(b_t) = r(b_t, \pi_t) + \mathop{\mathbb{E}}_{z_{t+1:T}} \left[ \sum_{\tau=t+1}^{T} r(b_\tau, \pi_\tau) \right]. \tag{1}$$

The action-value function is defined by executing action $a_t$ and then following policy $\pi$. The optimal value function may be computed using Bellman's principle of optimality,

$$V_t^{\pi^*}(b_t) = \max_{a_t} \{ r(b_t, a_t) + \mathop{\mathbb{E}}_{z_{t+1}|a_t, b_t} \left[ V_{t+1}^{\pi^*}(b_{t+1}) \right] \}. \tag{2}$$

The goal of the agent is to find the optimal policy $\pi^*$ that maximizes the value function.

## 3 Mathematical Analysis

Typically, it is infeasible to fully expand a Partially Observable Markov Decision Process (POMDP) tree due to the extensive computational resources and time required. To address this challenge, we propose two approaches. In the first approach, we propose a solver that selectively chooses a subset of the observations to branch from, while maintaining a full posterior belief at each node. This allows us to derive a hypothetical algorithm that directly uses our suggested deterministic bounds to choose which actions to take while exploring the tree. As in some scenarios computing a complete posterior belief may be too expensive, we suggest a second method that in addition to branching only a subset of the observations, selectively chooses a subset of the states at each encountered belief. Using the deterministic bounds at the root node for each action value allows us to certify the performance of a given policy at the root of the planning tree. Moreover, given a scenario in which an action exists whose lower bound surpasses all other actions' upper bound, the algorithm can identify the optimal action and stop further exploration. In contrast, existing state-of-the-art algorithms either do not provide any guarantee on the solution quality (e.g. POMCP Silver and Veness [2010]) or merely provide probabilistic guarantee on the solution (e.g. DESPOT Somani et al. [2013]). In the following section, we show how to use the deterministic bounds in conjunction with state-of-the-art algorithms to obtain performance guarantees. [1]

Both of the presented approaches diverge from many existing algorithms that rely on black-box prior, transition, and observation models. Instead, our method directly utilizes state and observation probability values to evaluate both the value function and the associated bounds. In return, we offer

---

[1] All proofs and derivations are deferred to the supplementary file.

anytime deterministic guarantees on the value function for the derived policy and establish bounds on its deviation from the value function of the optimal policy.

In this section we provide a mathematical quantification of the impact of using a solver that only considers a small subset of the theoretical tree branches and a subset of states within each node. We begin by defining a simplified POMDP, which is a reduced complexity version of the original POMDP that abstracts or ignores certain states and observations. We then establish a connection between the simplified value function and its theoretical counterpart. Finally, we demonstrate a relationship between the simplified value function, obtained by following the best policy for the simplified problem, and the theoretically optimal value function.

We begin with general definitions of a simplified prior, transition and observation models,

$$\bar{b}_0(x) \triangleq \begin{cases} b_0(x) & , \ x \in \bar{\mathcal{X}}_0 \\ 0 & , \ otherwise \end{cases} \tag{3}$$

$$\bar{\mathbb{P}}(x_{t+1} \mid x_t, a_t) \triangleq \begin{cases} \mathbb{P}(x_{t+1} \mid x_t, a_t) & , \ x_{t+1} \in \bar{\mathcal{X}}(H_{t+1}^-) \\ 0 & , \ otherwise \end{cases} \tag{4}$$

$$\bar{\mathbb{P}}(z_t \mid x_t) \triangleq \begin{cases} \mathbb{P}(z_t \mid x_t) & , \ z_t \in \bar{\mathcal{Z}}(H_t) \\ 0 & , \ otherwise \end{cases} \tag{5}$$

where $\bar{\mathcal{X}}(H_{t+1}^-) \subseteq \mathcal{X}$ and $\bar{\mathcal{Z}}(H_t) \subseteq \mathcal{Z}$ may be chosen arbitrarily, e.g. by sampling or choosing a fixed subset a-priori, as the derivations of the bounds are independent of the subset choice. Note that the simplified prior, transition and observation models are unnormalized and thus do not represent a valid distribution function. For the rest of the sequel we will drop the explicit dependence on the history, and denote $\bar{\mathcal{X}}(H_{t+1}^-) \equiv \bar{\mathcal{X}}$, $\bar{\mathcal{Z}}(H_t) \equiv \bar{\mathcal{Z}}$. Using the simplified models, we define the simplified belief $\bar{b}_{t+1}$. The simplified belief is updated using the simplified belief update equation, which is a modification of the standard belief update equation that replaces the usual models with the simplified ones. More precisely,

$$\bar{b}_{t+1}(x_{t+1}) \triangleq \begin{cases} \dfrac{\bar{\mathbb{P}}(z_{t+1} \mid x_{t+1}) \sum_{x_t} \bar{\mathbb{P}}(x_{t+1} \mid x_t, \pi_t) \bar{\mathbb{P}}(x_t \mid H_t)}{\bar{\mathbb{P}}(z_{t+1} \mid H_{t+1}^-)} & , \ \bar{\mathbb{P}}(z_{t+1} \mid H_{t+1}^-) \neq 0 \\ 0 & , \ otherwise \end{cases} \tag{6}$$

where $\bar{b}_{t+1}(x) \equiv \bar{\mathbb{P}}(x_{t+1} \mid H_{t+1})$, and $\bar{\mathbb{P}}(z_{t+1} \mid H_{t+1}^-) \triangleq \bar{\mathbb{P}}(z_{t+1} \mid x_{t+1}) \sum_{x_t} \bar{\mathbb{P}}(x_{t+1} \mid x_t, \pi_t) \bar{\mathbb{P}}(x_t \mid H_t)$. Note that a simplified belief cannot depend on an observation that is not part of the simplified observation set and is considered undefined. Last we define a simplified value function,

$$\bar{V}^\pi(\bar{b}_t) \triangleq r(\bar{b}_t, \pi_t) + \bar{\mathbb{E}}\left[\bar{V}(b_t)\right] \tag{7}$$

$$= \sum_{x_t} \bar{b}(x_t) r(x_t, \pi_t) + \sum_{z_t} \bar{\mathbb{P}}(z_{t+1} \mid H_{t+1}^-) \bar{V}(\bar{b}(z_{t+1})), \tag{8}$$

where the simplified expectation operator, $\bar{\mathbb{E}}[\cdot]$, is taken with respect to the unnormalized likelihood $\bar{\mathbb{P}}(z_{t+1} \mid H_{t+1}^-)$.

## 3.1 Simplified observation space

We first analyze the performance guarantees of a simplified observation space, while assuming a complete belief update at each belief state, i.e., $\bar{\mathcal{X}} \equiv \mathcal{X}$. The following theorem describes the guarantees of the observation-simplified value function with respect to its theoretical value,

**Theorem 1.** *Let $b_t$ belief state at time $t$, and $T$ be the last time step of the POMDP. Let $V^\pi(b_t)$ be the theoretical value function by following a policy $\pi$, and let $\bar{V}^\pi(b_t)$ be the simplified value function, as defined in* (18)*, by following the same policy. Then, for any policy $\pi$, the difference between the theoretical and simplified value functions is bounded as follows,*

$$\left|V^\pi(b_t) - \bar{V}^\pi(b_t)\right| \leq \mathcal{R}_{\max} \sum_{\tau=t+1}^{T} \left[1 - \sum_{z_{t+1:\tau}} \sum_{x_{t:\tau}} b(x_t) \prod_{k=t+1}^{\tau} \bar{\mathbb{P}}(z_k \mid x_k) \mathbb{P}(x_k \mid x_{k-1}, \pi_{k-1})\right] \triangleq \epsilon_z^\pi(b_t), \tag{9}$$

where we use a subscript of $(z)$ in $\epsilon_z^\pi(b_t)$ to denote observation-only simplification. Importantly, the bound only contains terms which depend on observations that are within the simplified space, $z \in \bar{\mathcal{Z}}$. This is an essential property of the bound, as it is a value that can easily be calculated during the planning process and provides a certification of the policy quality at any given node along the tree. Furthermore, it is apparent from (20) that as the number of observations included in the simplified set, $\bar{\mathcal{Z}}$, increases, the value of $\epsilon_z^\pi(b_t)$ consequently diminishes,

$$\sum_{z_{1:\tau}} \sum_{x_{0:\tau}} b(x_0) \prod_{k=1}^{\tau} \bar{\mathbb{P}}(z_k \mid x_k) \mathbb{P}(x_k \mid x_{k-1}, \pi_{k-1}) \xrightarrow{\bar{\mathcal{Z}} \to \mathcal{Z}} 1$$

leading to a convergence towards the theoretical value function, i.e. $\epsilon_z^\pi(b_t) \to 0$.

Theorem 3 provides both lower and upper bounds for the theoretical value function, assuming a fixed policy. Using this theorem, we can derive upper and lower bounds for any policy, including the optimal one. This is achieved by applying the Bellman optimality operator to the upper bound in a repeated manner, instead of the estimated value function; In the context of tree search algorithms, our algorithm explores only a subset of the decision tree due to pruned observations. However, at every belief node encountered during this exploration, all potential actions are expanded. The action-value of these expanded actions is bounded using the Upper Deterministic Bound, which we now define as

$$\text{UDB}^\pi(b_t, a_t) \triangleq \bar{Q}^\pi(b_t, a_t) + \epsilon_z^\pi(b_t, a_t) = r(b_t, a_t) + \bar{\mathbb{E}}_{z_{t+1}}[\bar{V}^\pi(b_{t+1})] + \epsilon_z^\pi(b_t, a_t), \qquad (10)$$

where the action-dependent bound on the value difference, $\epsilon_z^\pi(b_t, a_t)$, is the bound of taking action $a_t$ in belief $b_t$ and following policy $\pi$ thereafter,

$$\epsilon_z^\pi(b_t, a_t) \triangleq \mathcal{R}_{\max} \sum_{\tau=t+1}^{T} \Big[ 1 - \sum_{z_{t+1:\tau}} \sum_{x_{t:\tau}} b(x_t) \bar{\mathbb{P}}(z_{t+1} \mid x_{t+1}) \mathbb{P}(x_{t+1} \mid x_t, a_t) \cdot \qquad (11)$$
$$\prod_{k=t+2}^{\tau} \bar{\mathbb{P}}(z_k \mid x_k) \mathbb{P}(x_k \mid x_{k-1}, \pi_{k-1}) \Big].$$

In the event that no subsequent observations are chosen for a given history, the value of $\bar{Q}^\pi(b_t, a_t)$ simplifies to the immediate reward plus an upper bound for any subsequent policy, given by $\mathcal{R}_{\max} \cdot (T - t - 1)$.

Using UDB we define the action selection criteria according to

$$a_t = \arg\max_{a_t \in \mathcal{A}} [\text{UDB}^\pi(b_t, a_t)] = \arg\max_{a_t \in \mathcal{A}} [\bar{Q}^\pi(b_t, a_t) + \epsilon_z^\pi(b_t, a_t)]. \qquad (12)$$

Moreover, the optimal value function can be bounded as follows,

**Lemma 1.** *The optimal value function can be bounded as*

$$V^{\pi*}(b_t) \leq \text{UDB}^\pi(b_t), \qquad (13)$$

*where the policy $\pi$ is determined according to Bellman optimality over the UDB, i.e.*

$$\text{UDB}^\pi(b_t) \triangleq \max_{a_t \in \mathcal{A}} [\bar{Q}^\pi(b_t, a_t) + \epsilon_z^\pi(b_t, a_t)] \qquad (14)$$
$$= \max_{a_t \in \mathcal{A}} [r(b_t, a_t) + \bar{\mathbb{E}}_{z_{t+1}|b_t, a_t}[\bar{V}^\pi(b_{t+1})] + \epsilon_z^\pi(b_t, a_t)]. \qquad (15)$$

**Corollary 1.1.** *By utilizing Lemma 1 and the exploration criteria defined in* (65)*, an increasing number of explored belief nodes guarantees convergence to the optimal value function.*

Notably, UDB does not require a complete recovery of the posterior branches to yield an optimal policy. Each action-value is bounded by a specific lower and upper bound, which can be represented as an interval enclosing the theoretical value. When the bound intervals of two candidate actions do not overlap, one can clearly discern which action is suboptimal, rendering its subtree redundant for further exploration. This distinction sets UDB apart from current state-of-the-art online POMDP algorithms. In those methods, any finite-time stopping condition fails to ensure optimality since the bounds used are either heuristic or probabilistic in nature.

## 3.2 Simplified State and Observation Spaces

In certain scenarios, the complete evaluation of posterior beliefs during the planning stage may pose significant computational challenges. To tackle this issue, we propose the use of a simplified state space in addition to the simplified observation space considered thus far. Specifically, we derive deterministic guarantees of the value function that allow for the selection of a subset from both the states and observations.

**Theorem 2.** *Let $b_0$ and $\bar{b}_0$ be the theoretical and simplified belief states, respectively, at time $t = 0$, and $T$ be the last time step of the POMDP. Let $V^\pi(b_0)$ be the theoretical value function by following a policy $\pi$, and let $\bar{V}^\pi(\bar{b}_0)$ be the simplified value function by following the same policy, as defined in (18). Then, for any policy $\pi$, the difference between the theoretical and simplified value functions is bounded as follows,*

$$\left| V^\pi(b_0) - \bar{V}^\pi(\bar{b}_0) \right| \leq \mathcal{R}_{\max} \left[ 1 - \sum_x \bar{b}_0(x) \right] + \mathcal{R}_{\max} \sum_{\tau=1}^{T} \left[ 1 - \overline{\mathbb{E}}_{z_{1:\tau}} \sum_x \bar{b}_\tau(x) \right] \triangleq \epsilon_{x,z}^\pi(b_0),$$

(16)

Similar to Theorem 3, $\epsilon_{x,z}^\pi(b_0)$ only contains probability values of instances from the simplified sets. However, this bound accounts for both the states and observations that are within the simplified spaces, $x, z \in \bar{\mathcal{X}}, \bar{\mathcal{Z}}$, which makes it a viable to compute at planning time. In contrast to Theorem 3, this bound can only be calculated at the root since it relies on the knowledge of the actual probability value of the prior, $b_0(x)$, for states $x \in \bar{\mathcal{X}}$, which are only available at the root. Furthermore, since now the belief density function at different belief nodes is not required, we demonstrate in the next section that a belief update can be avoided completely, which makes it suitable to attach guarantees to state-of-the-art algorithms.

The process of maintaining an upper bound for action-value function follows similarly to the one presented in the observation-only simplification subsection 3.1,

$$\epsilon_{x,z}^\pi(b_t, a_t) \triangleq \mathcal{R}_{\max} \left[ 1 - \sum_{x_{t:t+1}} b(x_t) \overline{\mathbb{P}}(x_{t+1} \mid x_t, a_t) \right] + \tag{17}$$

$$\mathcal{R}_{\max} \sum_{\tau=t+1}^{T} \left[ 1 - \sum_{z_{t+1:\tau}} \sum_{x_{t:\tau}} b(x_t) \overline{\mathbb{P}}(z_{t+1} \mid x_{t+1}) \overline{\mathbb{P}}(x_{t+1} \mid x_t, a_t) \prod_{k=t+2}^{\tau} \overline{\mathbb{P}}(z_k \mid x_k) \overline{\mathbb{P}}(x_k \mid x_{k-1}, \pi_{k-1}) \right].$$

In $\epsilon_{x,z}^\pi(b_t, a_t)$, the subscript of $(x, z)$ differentiates the bound of the state- and observation-simplification from the observation-only simplification bound.

Computing the bounds in practice is straightforward and can be done iteratively, which fits many online planning algorithms. As it requires the summation over probability values for every considered trajectory, $x_0, a_0, x_1, z_1, ..., a_{t-1}, x_t, z_t$, as shown in equations (17) and (11). In the subsequent section, we provide further details and illustrate how the bounds derived from the state-observation simplification can be used to derive deterministic guarantees on policies obtained from existing state-of-the-art algorithms.

## 4 Methods

---

**Algorithm 1** ALGORITHM-$\mathcal{A}$:

---

**function** SEARCH
1: **while** time permits **do**
2:    Generate states $x$ from $b_0$.
3:    $\tau_0 \leftarrow x$
4:    $\bar{\mathbb{P}}_0 \leftarrow b(x_0 \mid h_0)$
5:    **if** $\tau_0 \notin \tau(h)$ **then**
6:       $\mathbb{P}(h) \leftarrow \mathbb{P}(h) + \bar{\mathbb{P}}_0$
7:    **end if**
8:    SIMULATE$(h, D, \tau_0, \bar{\mathbb{P}}_0)$.
9: **end while**
10: **return**

**function** FWDUPDATE$(ha, haz, \tau_d, \bar{\mathbb{P}}_\tau, x')$
1: **if** $\tau_d \notin \tau(ha)$ **then**
2:    $\tau(ha) \leftarrow \tau(ha) \cup \{\tau_d\}$
3:    $\bar{R}(ha) \leftarrow \bar{R}(ha) + \bar{\mathbb{P}}_\tau \cdot r(x, a)$
4: **end if**
5: $\tau_d \leftarrow \tau_d \cup \{x'\}$
6: $\bar{\mathbb{P}}_\tau \leftarrow \bar{\mathbb{P}}_\tau \cdot Z_{z|x'} \cdot T_{x'|x,a}$
7: **if** $\tau_d \notin \tau(haz)$ **then**
8:    $\bar{\mathbb{P}}(haz) \leftarrow \bar{\mathbb{P}}(haz) + \bar{\mathbb{P}}_\tau$
9:    $\tau(haz) \leftarrow \tau(haz) \cup \{\tau_d\}$
10: **end if**
11: **return**

**function** SIMULATE$(h, d, \tau_d, \bar{\mathbb{P}}_d)$
1: **if** $d = 0$ **then**
2:    **return** $\delta(h) \leftarrow 0$
3: **end if**
4: Select action $a$.
5: Generate next states and observations, $x', z$.
6: $\tau_d, \bar{\mathbb{P}}_\tau \leftarrow$FWDUPDATE$(ha, haz, \tau_d, \bar{\mathbb{P}}_\tau, x')$
7: Select next observation $z$.
8: SIMULATE$(haz, d - 1, \tau_d, \bar{\mathbb{P}}_\tau)$
9: BWDUPDATE$(h, ha, d)$
10: **return**

**function** BWDUPDATE$(h, ha, d)$
1: $\epsilon(ha) = \gamma^{D-d}V_{\max}(\bar{\mathbb{P}}(h) - \bar{\mathbb{P}}(ha)) + \gamma^{D-d-1} \cdot V_{\max}(\bar{\mathbb{P}}(ha) - \sum_{z|ha} \bar{\mathbb{P}}(haz))$
2: $U(ha) = \bar{R}(ha) + \gamma\sum_{z|ha} U(haz) + \epsilon(ha)$
3: $L(ha) = \bar{R}(ha) + \gamma\sum_{z|ha} L(haz) - \epsilon(ha)$
4: $U(h) \leftarrow \max_{a'}\{U(ha')\}$
5: $L(h) \leftarrow \max_{a'}\{L(ha')\}$
6: **return**

---

In this section we aim to describe how to fit our bounds to a general structure algorithm, named ALGORITHM $- \mathcal{A}$, which serves as an abstraction to many existing algorithms. To compute the deterministic bounds, we utilize Bellman's update and optimality criteria. This approach naturally fits dynamic programming approaches such as DESPOT [Ye et al., 2017] and AdaOPS [Wu et al., 2021]. However, it may also be attached with algorithms that rely on Monte-Carlo estimation, such as POMCP [Silver and Veness, 2010], by viewing the search tree as a policy tree.

While the analysis presented in section 3 is general and independent of the selection mechanism of the states or observations, we focus on sampling as a way to choose the simplified states at each belief node and the observations to branch from. Furthermore, the selection of the subspaces $\bar{\mathcal{X}}, \bar{\mathcal{Z}}$ need not be fixed, and may change over the course of time, similar to state-of-the-art algorithms, such as Hoerger and Kurniawati [2021], Silver and Veness [2010], Somani et al. [2013], Sunberg and Kochenderfer [2018], Wu et al. [2021]. Alternative selection methods may also be feasible, as sampling from the correct distribution is not required for the bounds to hold. Importantly, attaching our bounds to arbitrary exploration mechanism, such as in POMCP or DESPOT, leverages the derivations and convergence guarantees shown in 3.2. Clearly, the deterministic bounds remain valid, but a convergence analysis depends on the characteristics of the specific ALGORITHM $- \mathcal{A}$ being used.

ALGORITHM $- \mathcal{A}$ is outlined in algorithm 1. For the clarity of exposition, we assume the following; at each iteration a single state particle is propagated from the root node to the leaf (line 2 of function SEARCH). The selection of the next state and observations are done by sampling from the observation and transition models (line 5), and each iteration ends with the full horizon of the POMDP (lines 2). However, none of these are a restriction of our approach and may be replaced with arbitrary number of particles, arbitrary state and observation selection mechanism and a single or multiple expansions of new belief nodes at each iteration.

To compute the UDB value, we require both the state trajectory, denoted as $\tau$, and its probability value, $\mathbb{P}_\tau$. We use the state trajectory as a mechanism to avoid duplicate summation of an already accounted for probability value and is utilized to ascertain its uniqueness at a belief node. The probability value, $\mathbb{P}_\tau$, is the likelihood of visiting a trajectory $\tau = \{x_0, a_0, x_1, z_1, \ldots, a_{t-1}, x_t, z_t\}$ and is calculated as the product of the prior, transition and observation likelihoods (line 6). If a trajectory was not seen before in a belief node, its reward value is multiplied by the trajectory likelihood, shown in 3. Each node maintains a cumulative sum of the likelihoods of all visited trajectories. This is then being

used to compute the upper and lower bounds, shown in lines 2. The bounds are computed in lines 1 represent the loss of holding only a subset of the sates in node $ha$ from the set in node $h$, plus the loss of having only a subset of the future states and observations, where $V_{\max}$ represent the maximum possible value function. A simple bound on the value function may be $V_{\max} = \mathcal{R}_{\max} \cdot (D - d - 1)$, but other more sophisticated bounds are also possible, as well as different values for lower and upper bounds.

The time complexity for each posterior node, primarily depends on the specific algorithm being used. In the case of dynamic programming methods, such as DESPOT and AdaOPS, there is a negligible added computational complexity detailed below. In the case of Monte Carlo methods, such as POMCP, the computational complexity is $O(|\mathcal{A}|)$ attributed mainly to the action-selection, while our approach adds another linear time complexity term, making it $O(|\mathcal{A}| + |\bar{\mathcal{Z}}|)$ due to the summation over the simplified observation space. During each iteration of the algorithm, an "IF" statement is used to determine whether a specific trajectory has already been encountered at the current node. This verification process can potentially result in an added linear complexity of $O(D)$, where $D$ represents the planning horizon. However, this overhead can be circumvented by assigning a unique ID value to each trajectory at the previous step and subsequently checking whether a pair, comprising the ID value and the new state, has already been visited. This approach reduces the overhead to an average time complexity of $O(1)$ by utilizing hash maps efficiently.

## 5   Experiments

In this section, we present the experimental results obtained by integrating deterministic bounds into a state-of-the-art algorithms namely, AR-DESPOT Somani et al. [2013] and POMCP Silver and Veness [2010] as a baseline. The primary objective of these experiments is to demonstrate the validity of our derived bounds, as presented in Theorem 2, and the corresponding algorithm outlined in Algorithm 1.

The implementation of our algorithm was carried out using the Julia programming language and evaluated through the Julia POMDPs package, provided by Egorov et al. [2017]. Although the library primarily focused on infinite horizon problems, we made the required modifications to extend its capabilities to accommodate finite-horizon POMDPs. The experiments were conducted on a computing platform consisting of an Intel(R) Core(TM) i7-7700 processor with 8 CPUs operating at 3.60GHz and 15.6 GHz. The

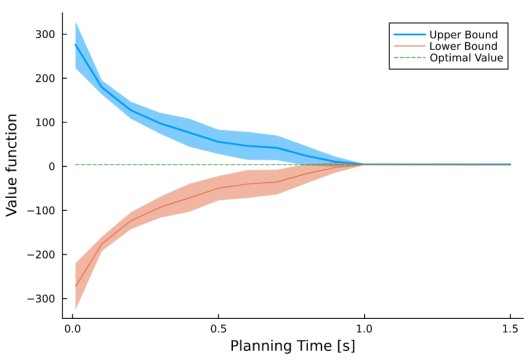

Figure 2: The graph illustrates the convergence of the deterministic bounds using Deterministically-Bounded-AR-DESPOT, with upper and lower bounds depicted alongside the optimal value obtained through exhaustive search.

selection of hyper-parameters for the POMCP and AR-DESPOT solvers, and further details about the POMDPs used for our experiments are detailed in the appendix.

In Figure 2, we demonstrate the empirical convergence of the deterministic bounds in relation to planning time. For these experiments, we focused on a toy example, Tiger POMDP Kaelbling et al. [1998]. By conducting an exhaustive search and computing a full posterior belief for each node, we derived the optimal value, depicted as a dashed green line in figure 2. The graph presents the convergence of the bounds calculated with Deterministically-Bounded AR-DESPOT (DB-DESPOT), to the optimal value. The mean values of the upper and lower bounds across 100 simulations are plotted, accompanied by the standard deviation, for time increments of $\Delta t = 0.1$ seconds.

Algorithms that provide probabilistic guarantees have a non-zero probability of taking a suboptimal action regardless of the planning time. In contrast, when given sufficient planning time, the deterministic bounds can certify the optimal action once the lower bound of an immediate action exceeds the upper bounds of all other actions. In our experiments, we observed that although the baseline algorithms, AR-DESPOT and POMCP, and the deterministically bounded algorithms, DB-DESPOT

and DB-POMCP, had access to the same tree and samples, AR-DESPOT and POMCP occasionally made incorrect decisions, resulting in degraded performance.

We evaluated the performance of both algorithms on different POMDPs, including the Tiger POMDP, Discrete Light Dark Sunberg and Kochenderfer [2018] and Baby POMDP. The corresponding results are summarized in Table 1. After each planning session the calculated best action is executed, the belief is updated according to the captured observation and a new planning session is invoked. The table reports the empirical mean and std of the cumulative rewards obtained by executing the calculated best actions based on 100 runs.

In the POMDPs with manageable state, action and observation sizes, both the DB-DESPOT and DB-POMCP algorithms frequently identified the optimal action within a 1-second time budget. It's important to highlight that these algorithms employ stochastic methods for belief tree exploration as in the baseline algorithms. Consequently, the outcomes between DB-DESPOT and DB-POMCP can vary, and there is no guarantee of consistently identifying the optimal action in every constrained-time planning session. An optimal action is determined when a lower bound of a given action is higher than any other upper bound; notably, this does not necessitates a construction of the entire planning tree and can be used as an early stopping mechanism for the optimal action. In the larger Laser Tag POMDP Somani et al. [2013], the DB-DESPOT and DB-POMCP did not outperform AR-DESPOT and POMCP. This discrepancy occurred because the planning time was insufficient to find an optimal action due to larger state, action and observation spaces of Laser Tag POMDP. While our proposed algorithms shows promise in more compact POMDPs, their performance in larger-scale problems, like the Laser Tag POMDP, especially when constrained by external time budgets, merits further investigation.

Table 1: Performance comparison with and without deterministic bounds.

| Algorithm | Tiger POMDP | Laser Tag | Discrete Light Dark | Baby POMDP |
|---|---|---|---|---|
| DB-DESPOT (ours) | $3.74\pm0.48$ | $-5.29\pm0.14$ | $-5.29\pm0.01$ | $-3.92\pm0.56$ |
| AR-DESPOT | $2.82\pm0.55$ | $-5.10\pm0.14$ | $-61.53\pm5.80$ | $-5.40\pm0.85$ |
| DB-POMCP (ours) | $3.01\pm0.21$ | $-3.97\pm0.24$ | $-3.70\pm0.82$ | $-4.48\pm0.57$ |
| POMCP | $2.18\pm0.76$ | $-3.92\pm0.27$ | $-4.51\pm1.15$ | $-5.39\pm0.63$ |

## 6 Conclusions

In this work, we presented a novel methodology aimed at offering anytime, deterministic guarantees for approximate POMDP solvers. These solvers strategically leverage a subset of the state and observation spaces to alleviate the computational overhead. Our key proposition elucidates a linkage between the optimal value function, which is inherently computationally intensive, and a more tractable approximation frequently employed in contemporary algorithms. In the first part of the paper, we derived the theoretical relationship between the use of a selective subset of observations in a planning tree. One contribution of this work is the formulation of an upper deterministic bound (UDB) which governs exploration within the belief space, and is theoretically guaranteed to converge to the optimal value. This approach, however, depends on a complete belief update at each node of the tree, a requirement that can be computationally infeasible in many practical POMDPs. In the second part, we address this computational challenge, by extending our derivations to account for both a subset of the observations and states. This extension increases the feasibility of our approach and allows it to be incorporated into existing state-of-the-art algorithms. We have outlined how our methodology can be integrated within these algorithms. To illustrate the practical utility of our derivations, we applied them to certify and improve the solution of AR-DESPOT and POMCP algorithms, a state-of-the-art solvers for POMDPs.

### 6.1 Limitations and future work

While the application of our deterministic bounds within the context of the AR-DESPOT algorithm showcased its potential, there are several challenges that need to be addressed in future work. Firstly, the convergence rate of our proposed method remains an area for further exploration. A detailed theoretical analysis on this aspect could provide valuable insights into the practicality of our approach

for larger, more complex domains. Secondly, the current use of the UDB is predominantly restricted to problems characterized by low state dimensionality. Extending the UDB to handle higher state dimensionality is a significant challenge due to the increased computational complexity involved. Finally, we consider the integration of the UDB with simplified state and observation spaces to be a promising future direction. This could potentially lead to a more comprehensive and efficient strategy for handling larger POMDPs, thereby further enhancing the applicability of our deterministic approach.

## Acknowledgments

This work was supported by the Israel Science Foundation (ISF) and by US NSF/US-Israel BSF.

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

# Appendices

## 7  Mathematical Analysis

We start by restating the definition of the simplified value function,

$$\bar{V}^{\pi}(\bar{b}_t) \triangleq r(\bar{b}_t, \pi_t) + \bar{\mathbb{E}}\left[\bar{V}(b_t)\right] \tag{18}$$

$$= \sum_{x_t} \bar{b}(x_t) r(x_t, \pi_t) + \sum_{z_t} \bar{\mathbb{P}}(z_{t+1} \mid H_{t+1}^-) \bar{V}(\bar{b}(z_{t+1})), \tag{19}$$

### 7.1  Theorem 1

**Theorem 3.** *Let $b_t$ belief state at time $t$, and $T$ be the last time step of the POMDP. Let $V^{\pi}(b_t)$ be the theoretical value function by following a policy $\pi$, and let $\bar{V}^{\pi}(b_t)$ be the simplified value function, as defined in (18), by following the same policy. Then, for any policy $\pi$, the difference between the theoretical and simplified value functions is bounded as follows,*

$$\left|V^{\pi}(b_t) - \bar{V}^{\pi}(b_t)\right| \leq \mathcal{R}_{\max} \sum_{\tau=t+1}^{T} \left[1 - \sum_{z_{t+1:\tau}} \sum_{x_{t:\tau}} b(x_t) \prod_{k=t+1}^{\tau} \bar{\mathbb{P}}(z_k \mid x_k) \mathbb{P}(x_k \mid x_{k-1}, \pi_{k-1})\right] \triangleq \epsilon_z^{\pi}(b_t). \tag{20}$$

*Proof.* For notational convenience, we derive the bounds for the value function by denoting the prior belief as $b_0$,

$$V_0^{\pi}(b_0) = \mathbb{E}_{z_{1:T}}\left[\sum_{t=0}^{T} r(b_t, a_t)\right] \tag{21}$$

applying the belief update equation,

$$V_0^{\pi}(b_0) = \sum_{z_{1:T}} \prod_{\tau=1}^{T} \mathbb{P}\left(z_{\tau} \mid H_{\tau}^-\right) \sum_{t=0}^{T} \left[\sum_{x_t} \frac{\mathbb{P}(z_t \mid x_t) \sum_{x_{t-1}} \mathbb{P}(x_t \mid x_{t-1}, \pi_{t-1}) b_{t-1}}{\mathbb{P}\left(z_t \mid H_t^-\right)} r(x_t, a_t)\right] \tag{22}$$

$$= \sum_{z_{1:T}} \prod_{\tau=1}^{T} \mathbb{P}\left(z_{\tau} \mid H_{\tau}^-\right) \sum_{t=0}^{T} \left[\sum_{x_{0:t}} \frac{\prod_{k=1}^{t} \mathbb{P}(z_k \mid x_k) \mathbb{P}(x_k \mid x_{k-1}, \pi_{k-1}) b(x_0)}{\prod_{\tau=1}^{t} \mathbb{P}\left(z_{\tau} \mid H_{\tau}^-\right)} r(x_t, a_t)\right] \tag{23}$$

$$= \sum_{t=0}^{T} \sum_{z_{1:T}} \sum_{x_{0:T}} \prod_{k=1}^{t} \mathbb{P}(z_k \mid x_k) \mathbb{P}(x_k \mid x_{k-1}, \pi_{k-1}) b(x_0) r(x_t, a_t) \tag{24}$$

which applies similarly to the simplified value function,

$$\bar{V}_0^{\pi}(b_0) = \sum_{t=0}^{T} \sum_{z_{1:T}} \sum_{x_{0:T}} \prod_{k=1}^{t} \bar{\mathbb{P}}(z_k \mid x_k) \mathbb{P}(x_k \mid x_{k-1}, \pi_{k-1}) b(x_0) r(x_t, a_t). \tag{25}$$

We begin the derivation by focusing on a single time step, $t$, and later generalize to the complete value function.

$$|\mathbb{E}_{z_{1:t}}[r(b_t)] - \overline{\mathbb{E}}_{z_{1:t}}[r(\bar{b}_t)]| \tag{26}$$

$$=|\sum_{z_{1:t}}\sum_{x_{0:t}}[\prod_{k=1}^{t}\mathbb{P}(z_k \mid x_k)\mathbb{P}(x_k \mid x_{k-1},\pi_{k-1})b(x_0)r(x_t) - \prod_{k'=1}^{t}\overline{\mathbb{P}}(z_{k'} \mid x_{k'})\mathbb{P}(x_{k'} \mid x_{k'-1},\pi_{k'-1})b(x_0)r(x_t)]| \tag{27}$$

$$\leq \sum_{z_{1:t}}\sum_{x_{0:t}}\left|r(x_t)\left[\prod_{k=1}^{t}\mathbb{P}(z_k \mid x_k)\mathbb{P}(x_k \mid x_{k-1},\pi_{k-1})b(x_0) - \prod_{k'=1}^{t}b(x_0)\overline{\mathbb{P}}(z_{k'} \mid x_{k'})\mathbb{P}(x_{k'} \mid x_{k'-1},\pi_{k'-1})\right]\right| \tag{28}$$

$$=\sum_{z_{1:t}}\sum_{x_{0:t}}|r(x_t)|\left[\prod_{k=1}^{t}\mathbb{P}(z_k \mid x_k)\mathbb{P}(x_k \mid x_{k-1},\pi_{k-1})b(x_0) - \prod_{k'=1}^{t}b(x_0)\overline{\mathbb{P}}(z_{k'} \mid x_{k'})\mathbb{P}(x_{k'} \mid x_{k'-1},\pi_{k'-1})\right] \tag{29}$$

where the second transition is due to triangle inequality, the third transition is equality by the construction, i.e. using the simplified observation models imply that the difference is nonnegative. We add and subtract, followed by rearranging terms,

$$=\sum_{z_{1:t}}\sum_{x_{0:t}}|r(x_t)| \tag{30}$$
$$[\prod_{k=1}^{t}\mathbb{P}(z_k,x_k \mid x_{k-1},\pi_{k-1})b(x_0) - \prod_{k=1}^{t-1}b(x_0)\overline{\mathbb{P}}(z_k,x_k \mid x_{k-1},\pi_{k-1})\mathbb{P}(z_t,x_t \mid x_{t-1},\pi_{t-1})$$
$$+\prod_{k=1}^{t-1}b(x_0)\overline{\mathbb{P}}(z_k,x_k \mid x_{k-1},\pi_{k-1})\mathbb{P}(z_t,x_t \mid x_{t-1},\pi_{t-1}) - \prod_{k'=1}^{t}b(x_0)\overline{\mathbb{P}}(z_{k'},x_{k'} \mid x_{k'-1},\pi_{k'-1})]$$

$$=\sum_{z_{1:t}}\sum_{x_{0:t}}|r(x_t)|\Big\{ \tag{31}$$
$$\mathbb{P}(z_t,x_t \mid x_{t-1},\pi_{t-1})\left[\prod_{k=1}^{t-1}\mathbb{P}(z_k,x_k \mid x_{k-1},\pi_{k-1})b(x_0) - \prod_{k=1}^{t-1}b(x_0)\overline{\mathbb{P}}(z_k,x_k \mid x_{k-1},\pi_{k-1})\right]$$
$$+\prod_{k=1}^{t-1}b(x_0)\overline{\mathbb{P}}(z_k,x_k \mid x_{k-1},\pi_{k-1})[\mathbb{P}(z_t,x_t \mid x_{t-1},\pi_{t-1}) - \overline{\mathbb{P}}(z_t,x_t \mid x_{t-1},\pi_{t-1})]\Big\}$$

applying Holder's inequality,

$$\leq \mathcal{R}_{\max}\sum_{z_{1:t}}\sum_{x_{0:t}}\mathbb{P}(z_t,x_t \mid x_{t-1},\pi_{t-1})\left[b(x_0)\prod_{k=1}^{t-1}\mathbb{P}(z_k,x_k \mid x_{k-1},\pi_{k-1}) - b(x_0)\prod_{k=1}^{t-1}\overline{\mathbb{P}}(z_k,x_k \mid x_{k-1},\pi_{k-1})\right] \tag{32}$$
$$+\mathcal{R}_{\max}\sum_{z_{1:t}}\sum_{x_{0:t}}\prod_{k=1}^{t-1}\overline{\mathbb{P}}(z_k,x_k \mid x_{k-1},\pi_{k-1})b(x_0)[\mathbb{P}(z_t,x_t \mid x_{t-1},\pi_{t-1}) - \overline{\mathbb{P}}(z_t,x_t \mid x_{t-1},\pi_{t-1})]$$

$$=\mathcal{R}_{\max}\sum_{z_{1:t}}\sum_{x_{0:t}}\mathbb{P}(z_t,x_t \mid x_{t-1},\pi_{t-1})\cdot \tag{33}$$
$$\left[b(x_0)\prod_{k=1}^{t-1}\mathbb{P}(z_k,x_k \mid x_{k-1},\pi_{k-1}) - b(x_0)\prod_{k=1}^{t-1}\overline{\mathbb{P}}(z_k,x_k \mid x_{k-1},\pi_{k-1})\right] + \mathcal{R}_{\max}\delta_t$$

$$=\mathcal{R}_{\max}\sum_{z_{1:t-1}}\sum_{x_{0:t-1}}\left[b(x_0)\prod_{k=1}^{t-1}\mathbb{P}(z_k,x_k \mid x_{k-1},\pi_{k-1}) - b(x_0)\prod_{k=1}^{t-1}\overline{\mathbb{P}}(z_k,x_k \mid x_{k-1},\pi_{k-1})\right] \tag{34}$$
$$+\mathcal{R}_{\max}\delta_t,$$

following similar steps recursively,

$$= \ldots = \mathcal{R}_{\max} \sum_{\tau=1}^{t} \delta_\tau. \tag{35}$$

Finally, applying similar steps for every time step $t \in [1, T]$ results in,

$$\left| V^\pi(b_t) - \bar{V}^\pi(b_t) \right| \leq \mathcal{R}_{\max} \sum_{t=1}^{T} \sum_{\tau=1}^{t} \delta_\tau \tag{36}$$

where,

$$\delta_\tau = \sum_{z_{1:\tau}} \sum_{x_{0:\tau}} \prod_{k=1}^{\tau-1} \bar{\mathbb{P}}(z_k, x_k \mid x_{k-1}, \pi_{k-1}) b(x_0) [\mathbb{P}(z_\tau, x_\tau \mid x_{\tau-1}, \pi_{\tau-1}) - \overline{\mathbb{P}}(z_\tau, x_\tau \mid x_{\tau-1}, \pi_{\tau-1})]$$

$$= \sum_{z_{1:\tau-1}} \sum_{x_{0:\tau-1}} \prod_{k=1}^{\tau-1} \bar{\mathbb{P}}(z_k, x_k \mid x_{k-1}, \pi_{k-1}) b(x_0) [1 - \sum_{z_\tau} \sum_{x_\tau} \overline{\mathbb{P}}(z_\tau, x_\tau \mid x_{\tau-1}, \pi_{\tau-1})] \tag{37}$$

plugging the term in (37) to (36) and expanding the terms results in the desired bound,

$$\left| V^\pi(b_t) - \bar{V}^\pi(b_t) \right| \leq \mathcal{R}_{\max} \sum_{\tau=t+1}^{T} \left[ 1 - \sum_{z_{t+1:\tau}} \sum_{x_{t:\tau}} b(x_t) \prod_{k=t+1}^{\tau} \overline{\mathbb{P}}(z_k \mid x_k) \mathbb{P}(x_k \mid x_{k-1}, \pi_{k-1}) \right] \tag{38}$$

which concludes our derivation. □

## 7.2 Lemma 1

**Lemma 2.** *The optimal value function can be bounded as*

$$V^{\pi*}(b_t) \leq \text{UDB}^\pi(b_t), \tag{39}$$

*where the policy $\pi$ is determined according to Bellman optimality over the UDB, i.e.*

$$\text{UDB}^\pi(b_t) \triangleq \max_{a_t \in \mathcal{A}} [\bar{Q}^\pi(b_t, a_t) + \epsilon_z^\pi(b_t, a_t)] \tag{40}$$

$$= \max_{a_t \in \mathcal{A}} [r(b_t, a_t) + \bar{\mathbb{E}}_{z_{t+1} \mid b_t, a_t} [\bar{V}^\pi(b_{t+1})] + \epsilon_z^\pi(b_t, a_t)]. \tag{41}$$

*Proof.* In the following, we prove by induction that applying the Bellman optimality operator on upper bounds to the value function in finite-horizon POMDPs will result in an upper bound on the optimal value function. The notations are the same as the ones presented in the main body of the paper. We restate some of the definitions from the paper for convenience.

The policy $\pi_t(b_t)$ determined by applying Bellman optimality at belief $b_t$, i.e.,

$$\pi_t(b_t) = \arg \max_{a_t \in \mathcal{A}} [\bar{Q}^\pi(b_t, a_t) + \epsilon_z^\pi(b_t, a_t)]. \tag{42}$$

As it will be needed in the following proof, we also define the value of a belief which includes in its history at least one observation out of the simplified set, e.g. $H_t = \{a_0, z_1, \ldots, z_k \notin \overline{\mathcal{Z}}, \ldots, z_t\}$ as being equal to zero. Explicitly,

$$\overline{V}_t^\pi(\mathbb{P}(x_t \mid a_0, z_1, \ldots, z_k \notin \overline{\mathcal{Z}}, \ldots, z_t)) \equiv 0 \quad \forall k \in [1, t]. \tag{43}$$

We also use the following simple bound,

$$V_{t,\max} \triangleq \mathcal{R}_{\max} \cdot (T - t - 1) \tag{44}$$

**Base case** $(t = T)$ - At the final time step $T$, for each belief we set the value function to be equal to the reward value at that belief state, $b_T$ and taking the action that maximizes the immediate reward,

$$\text{UDB}^\pi(b_T) = \max_{a_T} \{r(b_T, a_T) + \epsilon_z(b_T, a_T)\} \equiv \arg \max_{a_T} \{r(b_T, a_T)\} \tag{45}$$

which provides an upper bound for the optimal value function for the final time step, $V_T^\star(b_T) \leq \text{UDB}^\pi(b_T)$.

**Induction hypothesis** - Assume that for a given time step, $t$, for all belief states the following holds,

$$V_t^\star(b_t) \leq \text{UDB}^\pi(b_t). \tag{46}$$

**Induction step** - We will show that the hypothesis holds for time step $t - 1$. By the induction hypothesis,

$$V_t^\star(b_t) \leq \text{UDB}^\pi(b_t) \ \forall b_t, \tag{47}$$

thus,

$$Q^\star(b_{t-1}, a_{t-1}) = r(b_{t-1}, a_{t-1}) + \sum_{z_t \in \mathcal{Z}} \mathbb{P}\left(z_t \mid H_t^-\right) V_t^\star(b(z_t)) \tag{48}$$

$$\leq r(b_{t-1}, a_{t-1}) + \sum_{z_t \in \mathcal{Z}} \mathbb{P}\left(z_t \mid H_t^-\right) \text{UDB}^\pi(b(z_t)) \tag{49}$$

$$= r(b_{t-1}, a_{t-1}) + \sum_{z_t \in \mathcal{Z}} \mathbb{P}\left(z_t \mid H_t^-\right) \left[\overline{V}_t^\pi(b_t) + \epsilon_z^\pi(b_t)\right]. \tag{50}$$

For the following transition, we make use of lemma 3,

$$= r(b_{t-1}, a_{t-1}) + \overline{\mathbb{E}}_{z_t \mid b_{t-1}, a_{t-1}} \left[\overline{V}_t^\pi(b_t)\right] + \epsilon_z^\pi(b_{t-1}, a_{t-1}) \tag{51}$$

$$\equiv \text{UDB}^\pi(b_{t-1}, a_{t-1}). \tag{52}$$

Therefore, under the induction hypothesis, $Q_{t-1}^\star(b_{t-1}, a_{t-1}) \leq \text{UDB}^\pi(b_{t-1}, a_{t-1})$. Taking the maximum over all actions $a_t$,

$$\text{UDB}^\pi(b_{t-1}) = \max_{a_{t-1} \in \mathcal{A}} \left\{\text{UDB}^\pi(b_{t-1}, a_{t-1})\right\} \tag{53}$$

$$\geq \max_{a_{t-1} \in \mathcal{A}} \left\{Q_{t-1}^\star(b_{t-1}, a_{t-1})\right\} = V_{t-1}^\star(b_{t-1}),$$

which completes the induction step and the required proof. $\square$

**Lemma 3.** *Let $b_t$ denote a belief state and $\pi_t$ a policy at time $t$. Let $\overline{\mathbb{P}}(z_t \mid x_t)$ be the simplified observation model which represents the likelihood of observing $z_t$ given $x_t$. Then, the following terms are equivalent,*

$$\mathbb{E}_{z_t} \left[\overline{V}_t^\pi(b_t) + \epsilon_z^\pi(b_t)\right] = \overline{\mathbb{E}}_{z_t} \left[\overline{V}_t^\pi(b_t)\right] + \epsilon_z^\pi(b_{t-1}, a_{t-1}) \tag{54}$$

*Proof.*

$$\mathbb{E}_{z_t} \left[\overline{V}_t^\pi(b_t) + \epsilon_z^\pi(b_t)\right] = \tag{55}$$

$$\mathbb{E}_{z_t} \left[\overline{V}_t^\pi(b_t)\right] + \mathbb{E}_{z_t} \left[\mathcal{R}_{\max} \sum_{\tau=t+1}^{T} \left[1 - \sum_{z_{t+1:\tau}} \sum_{x_{t:\tau}} b_t \prod_{k=t+1}^{\tau} \overline{\mathbb{P}}(z_k \mid x_k) \mathbb{P}(x_k \mid x_{k-1}, \pi_{k-1})\right]\right] \tag{56}$$

focusing on the second summand,

$$\sum_{z_t \in \mathcal{Z}} \mathbb{P}\left(z_t \mid H_t^-\right) \mathcal{R}_{\max} \sum_{\tau=t+1}^{T} \left[1 - \sum_{z_{t+1:\tau}} \sum_{x_{t:\tau}} b_t \prod_{k=t+1}^{\tau} \overline{\mathbb{P}}(z_k \mid x_k) \mathbb{P}(x_k \mid x_{k-1}, \pi_{k-1})\right] \tag{57}$$

$$= \mathcal{R}_{\max} \sum_{\tau=t+1}^{T} \left[1 - \sum_{z_t} \mathbb{P}\left(z_t \mid H_t^-\right) \sum_{z_{t+1:\tau}} \sum_{x_{t:\tau}} b(x_t) \prod_{k=t+1}^{\tau} \overline{\mathbb{P}}(z_k \mid x_k) \mathbb{P}(x_k \mid x_{k-1}, \pi_{k-1})\right] \tag{58}$$

by marginalizing over $x_{t-1}$,

$$= \mathcal{R}_{\max} \sum_{\tau=t+1}^{T} [1 - \sum_{z_t} \mathbb{P}\left(z_t \mid H_t^-\right) \sum_{z_{t+1:\tau}} \sum_{x_{t-1:\tau}} \frac{\overline{\mathbb{P}}(z_t \mid x_t)\mathbb{P}(x_t \mid x_{t-1}, \pi_{t-1})b(x_{t-1})}{\mathbb{P}\left(z_t \mid H_t^-\right)} \cdot \tag{59}$$
$$\prod_{k=t+1}^{\tau} \overline{\mathbb{P}}(z_k \mid x_k)\mathbb{P}(x_k \mid x_{k-1}, \pi_{k-1})]$$

canceling out the denominator,

$$= \mathcal{R}_{\max} \sum_{\tau=t+1}^{T} [1 - \sum_{z_{t:\tau}} \sum_{x_{t-1:\tau}} \overline{\mathbb{P}}(z_t \mid x_t)\mathbb{P}(x_t \mid x_{t-1}, a_{t-1})b(x_{t-1}) \cdot \tag{60}$$
$$\prod_{k=t+1}^{\tau} \overline{\mathbb{P}}(z_k \mid x_k)\mathbb{P}(x_k \mid x_{k-1}, \pi_{k-1})] \equiv \epsilon_z^{\pi}(b_{t-1}, a_{t-1})$$

it is left to show that $\mathbb{E}_{z_t \mid b_{t-1}, a_{t-1}}\left[\overline{V}_t^{\pi}(b_t)\right] = \overline{\mathbb{E}}_{z_t \mid b_{t-1}, a_{t-1}}\left[\overline{V}_t^{\pi}(b_t)\right]$. By the definition of a value function of a belief not included in the simplified set, we have that,

$$\mathbb{E}_{z_t \mid b_{t-1}, a_{t-1}}\left[\overline{V}_t^{\pi}(b_t)\right] = \sum_{z_t \in \mathcal{Z}} \mathbb{P}\left(z_t \mid H_t^-\right) \overline{V}_t^{\pi}(b_t) \tag{61}$$

$$= \sum_{z_t \in \overline{\mathcal{Z}}} \mathbb{P}\left(z_t \mid H_t^-\right) \overline{V}_t^{\pi}(b_t) + \sum_{z_t \in \mathcal{Z} \backslash \overline{\mathcal{Z}}} \mathbb{P}\left(z_t \mid H_t^-\right) \overline{V}_t^{\pi}(b_t) \tag{62}$$

$$= \sum_{z_t \in \overline{\mathcal{Z}}} \overline{\mathbb{P}}\left(z_t \mid H_t^-\right) \cdot \overline{V}_t^{\pi}(b_t) + \sum_{z_t \in \mathcal{Z} \backslash \overline{\mathcal{Z}}} \mathbb{P}\left(z_t \mid H_t^-\right) \cdot 0 \tag{63}$$

$$= \overline{\mathbb{E}}_{z_t \mid b_{t-1}, a_{t-1}}\left[\overline{V}_t^{\pi}(b_t)\right], \tag{64}$$

which concludes the derivation. $\qquad\square$

## 7.3 Corollary 1.1

We restate the definition of UDB exploration criteria,

$$a_t = \arg\max_{a_t \in \mathcal{A}}[\text{UDB}^{\pi}(b_t, a_t)] = \arg\max_{a_t \in \mathcal{A}}[\bar{Q}^{\pi}(b_t, a_t) + \epsilon_z^{\pi}(b_t, a_t)]. \tag{65}$$

**Corollary 3.1.** *Using Lemma 2 and the exploration criteria defined in (65) guarantees convergence to the optimal value function.*

*Proof.* Let us define a sequence of bounds, $\text{UDB}_n^{\pi}(b_t)$ and a corresponding difference value between $\text{UDB}_n$ and the simplified value function,

$$\text{UDB}_n^{\pi}(b_t) - \bar{V}_n^{\pi}(b_t) = \epsilon_{n,z}^{\pi}(b_t), \tag{66}$$

where $n \in [0, |\mathcal{Z}|]$ corresponds to the number of unique observation instances within the simplified observation set, $\overline{\mathcal{Z}}_n$, and $|\mathcal{Z}|$ denotes the cardinality of the complete observation space. Additionally, for the clarity of the proof and notations, assume that by construction the simplified set is chosen such that $\overline{\mathcal{Z}}_n(H_t) \equiv \overline{\mathcal{Z}}_n$ remains identical for all time steps $t$ and history sequences, $H_t$ given $n$. By the definition of $\epsilon_{n,z}^{\pi}(b_t)$,

$$\epsilon_{n,z}^{\pi}(b_t) = \mathcal{R}_{\max} \sum_{\tau=t+1}^{T} \left[1 - \sum_{z_{t+1:\tau} \in \overline{\mathcal{Z}}_n} \sum_{x_{t:\tau}} b(x_t) \prod_{k=t+1}^{\tau} \overline{\mathbb{P}}(z_k \mid x_k)\mathbb{P}(x_k \mid x_{k-1}, \pi_{k-1})\right], \tag{67}$$

we have that $\epsilon_{n,z}^{\pi}(b_t) \to 0$ as $n \to |\mathcal{Z}|$, since

$$\sum_{z_{t+1:\tau} \in \overline{\mathcal{Z}}_n} \sum_{x_{t:\tau}} b(x_t) \prod_{k=t+1}^{\tau} \overline{\mathbb{P}}(z_k \mid x_k)\mathbb{P}(x_k \mid x_{k-1}, \pi_{k-1}) \to 1 \tag{68}$$

as more unique observation elements are added to the simplified observation space, $\overline{\mathcal{Z}}_n$, eventually recovering the entire support of the discrete observation distribution.

From lemma 2 we have that, for all $n \in [0, |\mathcal{Z}|]$ the following holds,

$$V^{\pi*}(b_t) \leq \text{UDB}_n^{\pi}(b_t) = \bar{V}_n^{\pi}(b_t) + \epsilon_{n,z}^{\pi}(b_t). \tag{69}$$

Additionally, from theorem 3 we have that,

$$\left| V^{\pi}(b_t) - \bar{V}_n^{\pi}(b_t) \right| \leq \epsilon_{n,z}^{\pi}(b_t), \tag{70}$$

for any policy $\pi$ and subset $\overline{\mathcal{Z}}_n \subseteq \mathcal{Z}$, thus,

$$\bar{V}_n^{\pi}(b_t) - \epsilon_{n,z}^{\pi}(b_t) \leq V^{\pi}(b_t) \leq V^{\pi*}(b_t) \leq \bar{V}_n^{\pi}(b_t) + \epsilon_{n,z}^{\pi}(b_t). \tag{71}$$

Since $\epsilon_{n,z}^{\pi}(b_t) \to 0$ as $n \to |\mathcal{Z}|$, and $|\mathcal{Z}|$ is finite, it is guaranteed that $\text{UDB}_n^{\pi}(b_t) \xrightarrow{n \to |\mathcal{Z}|} V^{\pi*}(b_t)$ which completes our proof. $\square$

Moreover, depending on the algorithm implementation, the number of iterations can be finite (e.g. by directly choosing actions and observations to minimize the bound). A stopping criteria can also be verified by calculating the difference between the upper and lower bounds. The optimal solution is obtained once the upper bound equals the lower bound.

## 8 Experiments

### 8.1 POMDP scenarios

We begin with a brief description of the Partially Observable Markov Decision Process (POMDP) scenarios implemented for the experiments. each scenario was bounded by a finite number of time steps used for every episode, where each action taken by the agent led to a decrement in the number of time steps left. After the allowable time steps ended, the simulation was reset to its initial state.

#### 8.1.1 Tiger POMDP

The Tiger is a classic POMDP problem Kaelbling et al. [1998], involves an agent making decisions between two doors, one concealing a tiger and the other a reward. The agent needs to choose among three actions, either open each one of the doors or listen to receive an observation about the tiger position. In our experiments, the POMDP was limited horizon of 5 steps. The problem consists of 3 actions, 2 observations and 2 states.

#### 8.1.2 Discrete Light Dark

Is an adaptation from Sunberg and Kochenderfer [2018]. In this setting the agent needs to travel on a 1D grid to reach a target location. The grid is divided into a dark region, which offers noisy observations, and a light region, which offers accurate localization observations. The agent receives a penalty for every step and a reward for reaching the target location. The key challenge is to balance between information gathering by traveling towards the light area, and moving towards the goal region.

#### 8.1.3 Laser Tag POMDP

In the Laser Tag problem, Somani et al. [2013], an agent has to navigate through a grid world, shoot and tag opponents by using a laser gun. The main goal is to tag as many opponents as possible within a given time frame. The grid is segmented into various sections that have varying visibility, characterized by obstacles that block the line of sight, and open areas. There are five possible actions,

moving in four cardinal directions (North, South, East, West) and shooting the laser. The observation space cardinality is $|\mathcal{Z}| \approx 1.5 \times 10^6$, which is described as a discretized normal distribution and reflect the distance measured by the laser. The states reflect the agent's current position and the opponents' positions. The agent receives a reward for tagging an opponent and a penalty for every movement, encouraging the agent to make strategic moves and shots.

### 8.1.4 Baby POMDP

The Baby POMDP is a classic problem that represents the scenario of a baby and a caregiver. The agent, playing the role of the caregiver, needs to infer the baby's needs based on its state, which can be either crying or quiet. The states in this problem represent the baby's needs, which could be hunger, discomfort or no need. The agent has three actions to choose from: feeding, changing the diaper, or doing nothing. The observations are binary, either the baby is crying or not. The crying observation does not uniquely identify the baby's state, as the baby may cry due to hunger or discomfort, which makes this a partially observable problem. The agent receives a reward when it correctly addresses the baby's needs and a penalty when the wrong action is taken.

## 8.2 Hyperparameters

The hyperparameters for both DB-DESPOT and AR-DESPOT algorithms were selected through a grid search. We explored an array of parameters for AR-DESPOT, choosing the highest-performing configuration. Specifically, the hyperparameter $K$ was varied across $\{10, 50, 500, 5000\}$, while $\lambda$ was evaluated at $\{0, 0.01, 0.1\}$. Similarly, DB-POMCP and POMCP were examined three different values for the exploration-exploitation weight, $c = \{0.1, 1.0, 10.0\}$ multiplied by $V_{max}$, which denotes an upper bound for the value function.

For the initialization of the upper and lower bounds used by the algorithms, we used the maximal reward, multiplied by the remaining time steps of the episode, $\mathcal{R}_{\max} \cdot (\mathcal{T} - t - 1)$.

Finally, we provide our algorithm implementation in *https://github.com/moranbar/Online-POMDP-Planning-with-Anytime-Deterministic-Guarantees*.

