# Online POMDP Planning with Anytime Deterministic Guarantees - Supplementary

This document provides supplementary material to Online POMDP Planning with Anytime Deterministic Guarantees [1] and should not be considered a self-contained document. Throughout this report, all notations and definitions are in compliance with

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

For upper and lower bounds used both by DB-DESPOT (which results in deterministic bounds) and AR-DESPOT (which result in probabilistic bounds); we used the maximal reward, multiplied by the remaining time steps of the episode, $\mathcal{R}_{\max} \cdot (\mathcal{T} - t - 1)$.

Finally, we provide our algorithm implementation in *[will be provided upon official publication of the paper]*.