# OpenReview forum: "Online POMDP Planning with Anytime Deterministic Guarantees"
_NeurIPS.cc/2023/Conference — NeurIPS 2023 poster_

### Official Review · Reviewer_3sFn · 2023-06-14

**Soundness:** 4 excellent
**Presentation:** 3 good
**Contribution:** 3 good
**Rating:** 6
**Confidence:** 4

**Summary:**

This paper provides theoretical bounds for POMDP planning algorithms and derive a new planning algorithms. The method applies to POMDP with discrete actions and observations spaces where the transition and observation distributions are known.

The derivation of the bound relies on considering a POMDP with only a subset of observations and states with unnormalized distributions for the transition and the observation.

The authors proceed to compute the difference in value for this simplified POMDP and the original problem for a given policy and then extend the derivation to the optimal policy in the case of limited observations. This step should address the fact that tree based solver have limited branching in the observations.

Then they update the bounds to address subsets of states. This part addresses the challenge that the belief update requires summing over all possible states which can be expensive.

The theoretical results are used to improved upon existing tree-based POMDP planners and the resulting algorithm is evaluated on well-known benchmark problems.


**Strengths:**

They are able to leverage the explicit transition model in a tree-based POMDP solver. They derive bounds using a creative idea of a simplified POMDP model ignoring the observations that are not in the planning tree.

The theoretical bounds seem easier to use than previous probabilistic bounds in AR-DESPOT. However, the computational tractability of the bounds seem to be very reliable on the sampling of the states.

They show competitive performance on standard POMDP problems against a state of the art algorithm.

The main approach is quite easy to follow and mostly well motivated.


**Weaknesses:**

Soundness / intuitive justification of the choice of the simplified model. Although the mathematical derivation of the bound seem correct, it would be nice to further motivate this choice.

The notation is sometimes confusing and could use extra clarification (see questions for specific examples). Equation (6), none of the distributions are normalized, yet the simplified belief is normalized.

If my understanding is correct, given a POMDP planning tree, and a subset of states per node, the bounds can be calculated and are deterministic guarantees. However, the construction of the tree itself is still probabilistic since sampling is used at two places: sampling the observations, and sampling the subset of states to use. I think it was a bit misleading initially, the bounds are deterministic, but the performance of the algorithm will still be probabilistic. The authors could clarify that aspect, and comment on whether the convergence properties change or would be the same as the underlying algorithm using the bounds.

The paragraph line 215 was a very clear explanation of how to perform the calculation of the bound and could come sooner.

There are some inaccuracies in the complexity discussion: “the Bellman update is linear in terms of time complexity”. This statement does not make sense.

It would have been very strong if the bounds had been added to POMCP or POMCPOW as well to show that the theoretical results indeed applies to different algorithms.

I did not understand the discussion about why DB-DESPOT did not perform better on the Laser tag problem.

In the supplementary material, it wouldn’t hurt to add a few extra steps between 4 and 7 to make it easier on the reader. Specifically between step (6) and (7).


Minor typos and formatting:
- In the background in the belief equation, take $P(z_t \mid x_t)$ out of the sum for easier mapping with Equation (6)
- Equation (6) should be $b_{t+1}(x_{t+1})$
- Equation (28) in supplementary material seems wrong.
- L208 A is outlined.
- L220 seeing->seen


**Questions:**

- Does $\bar{P}(x_t | H_t) = \bar{b_t}(x_t)$ ?

- What is $H_t^-$? The superscript – is not explained I think.

- Why normalizing the simplified belief?

- Can you clarify the dependency between the bound and the future observations? Why $\epsilon_z$?

- How is the subset of state determined to derive the bound in theorem 2?

- For theorem 2, does it have the property that if the subset of states tends towards the full state space then the bound goes to 0?

- What time horizon is used to compute the optimal value function in figure 2?

- What do you mean by “non overlapping bounds between actions at the decision time”?

- Is the computation of the bound more expensive than one Bellman backup?


**Limitations:**

Using black-box models for transition and observations can be easier to implement in practice.

The authors could be a bit more up front in the beginning of the paper about the finite horizon aspect, although it might not matter much in practice since the tree-based method have limited depth anyway.

The author mention the limitation regarding the computational complexity, however it was not very clear to me if this was a limitation of their method specifically or a general statement since they also mention that the added complexity is negligible compared to AR-DESPOT.

---

> ### Author Rebuttal · Authors · 2023-08-08
>
> - **Probabilistic performance** - The reviewer is correct that the algorithmic examples shown in this paper lead to probabilistic performance with deterministic guarantees. In practice, the performance can become deterministic with a deterministic selection of the states and observations. However, this is not explored in this paper and may be an exciting topic for a follow-up paper.
>
> - **Complexity analysis of Monte-Carlo algorithms with our bounds** - Lines (233-238) refers to the Bellman-like update of our bounds using online Monte-Carlo algorithms such as POMCP. Since the updates at each posterior node in a POMCP algorithm are at most $O(|A|)$, our bounds add another term to the linear complexity to an overall of $O(|A|+|\bar{\mathcal{Z}}|)$, where $|\bar{\mathcal{Z}}|$ is the simplified observation set. We agree with the reviewer that the phrasing may be misleading; this point will be clarified in the final version of the paper.
>
> - **Additional algorithms** - As discussed in the paper, our bounds are also applicable to the POMCP algorithm; we will provide a pseudo-code of our approach to POMCP, which will give deterministic guarantees on the result of the algorithm.
>
> - **Convergence of algorithm** - The bounds discussed in this paper provide the worst-case gap between the simplified and the theoretical value function. When an action $a$ has it lower bound surpasses the upper bounds of all other actions (i.e., their bounds interval do not overlap), it is theoretically guaranteed that action $a$ is optimal. However, before that, when given little time for planning,  the optimal action is unknown. Since multiple actions may be optimal, selecting one can be made based on different heuristics to break the tie. For instance, heuristics may include the action with the highest lower bound, the highest upper bound, or the action with the highest number of visitations. In such scenarios, the selected action is not guaranteed to be optimal (as with LaserTag) but is guaranteed to have a bounded difference from the optimal value. In this example, the AR-DESPOT heuristic was empirically shown to perform better.
>
> - **Equation (28)** - The proof of Lemma 1 in the supplementary refers to the case where only observations are simplified. In that case, $\epsilon^\pi (b_T, a_T) \triangleq 0$ since no observations are involved, and the immediate reward is computed entirely. A further clarification will be added in the final manuscript.
>
> - **Further answers, corresponding to the order of the Questions section:**
>   + $\bar{\mathbb{P}}(x_t\mid H_t) \triangleq \bar{b}_t(x_t)$ as shown in equation (6).
>   + $H_t^-$ is the propagated belief, as shown in line 82.
>   + The subscript $z$ in $\epsilon^{\pi}_z(\cdot,\cdot)$ was mainly to disambiguate the action dependency of that function from $\epsilon^{\pi}(\cdot)$.
>   + In theorem 2, the subset can be chosen arbitrarily. For instance, using a POMCP-like approach, it could be determined online by sampling or offline before planning. The choice does not affect the bounds' correctness but may affect how loose the bounds are.
>   + Yes, as the full subset of states tends to the complete state space, the bound will converge to zero, thus recovering the theoretical value function.
>   + The horizon in that experiment was limited to five steps.
>   + As mentioned earlier, each action has different upper and lower bounds. When the interval determined by the lower- and upper-bound for action $a$ does not intersect with the bound interval of any other action, we say that the bounds of action $a$ do not overlap. For instance, when the lower bound of action $a$ is higher than the upper bound of all other actions.
>   + Generally no. Compared to the complete Bellman update, our algorithm improves the computational complexity by performing a Bellman update with approximations. In practice, algorithms such as the original AR-DESPOT algorithm perform approximate Bellman-update in the back-propagation phase, which costs $O(|A|+|\bar{\mathcal{Z}|})$. DB-DESPOT(ours) performs similar calculations and thus adds no additional computational cost.

---

> > ### Comment · Reviewer_3sFn · 2023-08-10
> > **Thank you for the detailed clarifications**
> >
> > I acknowledge reading the rebuttal and thank the authors for addressing my comments.
> >
> > Regarding the complexity analysis, since it comes up in the other review as well I encourage the authors to add this part tot he main paper.
> >
> > I think there is still some work that needs to be done on the $\epsilon$ notation, the interpretation of the subscripts in relation with equation (9) is a bit confusing.
> >
> > I had misunderstood the action-dependent bound part and the comments help the understanding.
> >
> > I understand the time constraint but I do believe adding POMCP(OW) rows in table 1 could be very valuable.

---

> > > ### Author Response · Authors · 2023-08-14
> > >
> > > We thank the reviewer for the suggestions on improving the accessibility of the text and notation; these will be considered in the final manuscript.
> > >
> > > We have also applied our formulation of deterministic bounds for POMCP, as suggested by the reviewer. The results for POMCP, and DB-POMCP (our method) are provided below, these will be updated in the final manuscript, in Table 1.
> > >
> > > | Algorithm | Tiger POMDP | Laser Tag | Discrete Light Dark | Baby POMDP |
> > > |---------|---------|---------|---------|---------|
> > > | **DB-POMCP (ours)**  | 3.01±0.21  | -3.97±0.24  | -3.70±0.82  | -4.48±0.57  |
> > > | **POMCP**  | 2.18±0.76  | -3.92±0.27  | -4.51±1.15  | -5.39±0.63  |

---

> > > > ### Comment · Reviewer_3sFn · 2023-08-15
> > > >
> > > > Great, those extra results make a very convincing point for the applicability of the theory.

---

### Official Review · Reviewer_T4b6 · 2023-07-06

**Soundness:** 2 fair
**Presentation:** 2 fair
**Contribution:** 3 good
**Rating:** 5
**Confidence:** 1

**Summary:**

The paper derives deterministic upper and lower bounds for POMDP. It also proposes a relationship between the computationally intensive optimal value function and a more accessible, simplified approximation.

**Strengths:**

This paper provides a strong theoretical analysis

**Weaknesses:**

The empirical results are limited

**Questions:**

N/A

---

### Official Review · Reviewer_GBcp · 2023-07-09

**Soundness:** 2 fair
**Presentation:** 2 fair
**Contribution:** 2 fair
**Rating:** 6
**Confidence:** 3

**Summary:**

This paper studies online planning in partially observable Markov decision processes (POMDP). POMDP is known to be a difficult problem, and this paper focuses on how to get anytime deterministic guarantees. This paper proposes to consider a simplified version of the problem by using only a subset of observation space and a subset of state space. They prove the gap between the value functions of an algorithm $\pi$ in the original problem and in the simplified problem. The gap mainly depends on how many states or observations are dropped in the simplified problem.

**Strengths:**

1. This paper studies an important and difficult problem, online planning in POMDP.

2. This paper provides an idea to simplified the problem by reducing the state space and observation space, which seems interesting to me.

**Weaknesses:**

1. The nontriviality and contribution of the results seem limited. Theorem 1 and Theorem 2 basically shows that the value functions of the algorithm $\pi$ in the original problem and in the simplified problem depends on how many states or observations are dropped in the subset that is used. This seems trivial to me. The relations shown in Theorem 1 and Theorem 2 seem also trivial. The second term in the bracket basically shows the probability that the chosen observations can be received, and then the gap between the value functions depends on the probability of the other observations that cannot be received, i.e., one minus the previous probability. Moreover, is the selection of the subset of states and observations more important and interesting, compared to showing the impact on the gap of value functions of dropped states and observations?

2. The presentation needs to be improved. See my questions below. Moreover, according to the statement in line 88, the policy considered is a deterministic policy, why the random policy is not considered? Around line 105, some intuition about why "This will allow us to certify the performance of a given policy at the root of the planning tree" will be easier for me to understand. According to line 121, the subsets could be chosen arbitrarily. But then, how could you guarantee that the probabilities in equations (3), (4) and (5) are well-defined?

**Questions:**

1. What is $R_{max}$ in equation (9) and (16)? It seems not defined in the paper.

2. Why is there an algorithm 1? It seems not mentioned or used in the paper.

3. What is the difference between $\epsilon^{\pi}(b_t)$ in equation (9) and $\epsilon_z^{\pi}(b_t,a_t)$ defined in equation (11)? Is $\epsilon^{\pi}(b_t)$ just a special case of $\epsilon_z^{\pi}(b_t,a_t)$ when the action $a_t = \pi_t$?

**Limitations:**

See weaknesses and questions above.

---

> ### Author Rebuttal · Authors · 2023-08-08
>
> - **Contribution** - We believe that the theoretical contribution of an algorithm with deterministic bounds with respect to the optimal value is significant and novel. To the authors' knowledge, no online planning algorithm exists for POMDPs with deterministic guarantees. If the reviewer finds the resulting bounds clear and simple, we consider this a benefit of the suggested method, contributing to its accessibility.
> In addition to the theoretical contribution, we suggest a general blueprint of an algorithm that suits in structure to multiple state-of-the-art algorithms, for instance, DESPOT and POMCP and their variants (e.g., AR-DESPOT [1], AdaOPS [2], POMCPOW [3], PFT-DPW [3]), and discuss how our bounds can be attached to many of those algorithms to provide deterministic guarantees. Crucially, these bounds are not merely theoretical but are computed online, with marginal effect on the planning efficiency.
> In the first part of this paper, we use sampling of the observations from their respective distribution; this is common in current state-of-the-art algorithms and was empirically shown to perform well in practice. In the second part, we suggest accompanying bounds to a given algorithm, e.g., AR-DESPOT, which depends on the algorithm to select the states and observations. Using the bounds to efficiently choose the subset of states and observations is another interesting direction and is deferred for future research.
>
> - **Deterministic policy** - In this paper, we have decided to address the case of a deterministic policy. We believe the bounds can be easily generalized to the case of stochastic policy using similar derivations, but this is not shown in this paper.
>
> - **Certify performance at the root** - Theorem 2, which examines the discrepancy between the theoretical value function and its simplified version, offers deterministic bounds at the root node, which are computed in practice. In contrast to algorithms that do not provide any finite time guarantees (e.g., POMCP) or algorithms that only provide probabilistic guarantees (e.g., DESPOT), we apply Theorem 2 in practice to determine both the upper and lower bounds of the value functions for every action selection at the root node of the tree. Furthermore, envision a scenario in which an action exists whose lower bound surpasses all other actions' upper bound. In that case, we can identify the optimal action, thus certifying the solution's optimality.
>
> - **Choosing the state-observation subsets** - The subsets may be chosen arbitrarily, while the simplified models in equations (3-5) are not intended to represent valid distribution functions, as stated explicitly in lines (121-136). Instead, they are merely functions approximating the original models, which relax some of the computational burden when exploring the tree.
>
> - **$R_{max}$** - Is the maximum possible reward value. We appreciate the reviewer noticing the missing definition; we will add it in the final version of the paper.
>
> - **Algorithm 1** - Complements Algorithm 2, which describes the recursive procedure. It is required to have a complete overview of the algorithm as it generates some of the required values that are transferred to Algorithm 2. We will add a clarification in the final version.
>
> - **$\epsilon^{\pi}(b_t), \epsilon^{\pi}_z(b_t, a_t)$** - Indeed as the author noted, $\epsilon^{\pi}(b_t)=\epsilon^{\pi}_z(b_t, \pi_t)$.
>
> [1] Nan Ye, Adhiraj Somani, David Hsu, and Wee Sun Lee. Despot: Online pomdp planning with regularization. JAIR, 58:231–266, 2017.
>
> [2] Chenyang Wu, Guoyu Yang, Zongzhang Zhang, Yang Yu, Dong Li, Wulong Liu, and Jianye Hao. Adaptive online packing-guided search for pomdps. In M. Ranzato, A. Beygelzimer, Y. Dauphin, P.S. Liang, and J. Wortman Vaughan, editors, Advances in Neural Information Processing Systems (NIPS), volume 34, pages 28419–28430. Curran Associates, Inc., 2021.
>
> [3] Zachary Sunberg and Mykel Kochenderfer. Online algorithms for pomdps with continuous state, action, and observation spaces. In Proceedings of the International Conference on Automated Planning and Scheduling, volume 28, 2018.

---

> > ### Comment · Reviewer_GBcp · 2023-08-17
> >
> > Thanks for the response. I strongly recommend you to add some clarification for those notations and especially the meaning of algorithm 1. It is not clear where you discuss it and how algorithm 2 uses it. And the names of both algorithm 1 and algorithm 2 are Algorithm-$\mathcal{A}$, this is quite misleading in the context. These confused me for understanding the method in sec. 4.

---

> > > ### Author Response · Authors · 2023-08-18
> > >
> > > We appreciate the constructive feedback to enhance the clarity of our paper. We agree with the points raised and will address them, along with the other suggestions from the first review, in the revised manuscript.

---

### Official Review · Reviewer_JfUf · 2023-07-27

**Soundness:** 2 fair
**Presentation:** 1 poor
**Contribution:** 3 good
**Rating:** 6
**Confidence:** 3

**Summary:**

This submission develops a method for POMDP planning based on a notion of limiting the states and observations that have been and will be encountered during the tree search.  Two cases are developed, one where only observations are limited, and one where both states and observations are limited.

In both cases, the theory develops the notion of a value associated with this limited setting, and shows that the limited value has a well-defined absolute-error bound w.r.t. the real `unlimited` value function.  Such bounds are then used to choose actions in an UCB-like fashion, which results in the final planning method.

The simplified POMDP formalism is quite interesting, and explained clearly enough.  However, it might benefit from aditional discusssion about the assumptions that the limited observation and state sets need to satisfy, e.g., it seems like the limites state set must contain at least one state that is supported by the non-limited initial belief.  It also seems like the observation and state sets may not always be chosen freely and independently, because if the limited observation set allows for limited histories whose beliefs **only** support states that don't belong to the limited state set, then the limited version of belief updates shown in equation (6) breaks with a division by 0.

**Strengths:**

The topic addressed by the authors is interesting and novel, and a theoretically driven approach is highly appreciated.  With significant improvements in the presentation and evaluation, this work will become much stronger.

**Weaknesses:**

The inconsistency of the presentation and notation is a significant drawback to comprehension. The first few sections of the document are well written and clear, however the last few sections seem rushed and with a significant drop in discussion and analysis.  The general impression I get is that the writing of the final parts was rushed due to lack of time.  The mathematical formalisms are for the most part clear, but there are some instances of typos, inconsistent notation, and undefined terms that should be addressed.  I invite the authors to carefully review the math to make sure these issues are addresses.  The algorithm description and discussion in section (4) also may use improvements and clarifications.

The exact significance of the theoretical erorr bounds for any given limited set Z or X is unclear, as it seems like they may be as large as the real values themselves.  The main theorems of this paper seem to only guarantee optimality in the limit as the limited sets Z and X grow to cover the respective original sets.

The evaluation setting seems a bit questionable and confusing;  It seems like the main advantage of the proposed method would be in tackling complicated POMDPs that have many possible states and observations.  Instead, the evaluation is performed on the Tiger problem, which contains only 2 possible states and 2 possible observations.  The tiger problem contains complexities that are related to the estimation of values in the highly stochastic and uncertain setting, but in terms of state and observation spaces, it is truly minimal, to the point that it's unclear how a method that is designed around a notion of "ignoring" certain states and observations could possibly work well or even at all.  Any clarification would be greatly appreciated.  In addition, a broader discussion about sizes of these problems, and how they related to the possibility of limiting the observation and state spaces would greatly help make the evaluation section more convincing.

In addition to the above, it seems like the evaluation could be performed on larger problems, since for tiger, discrete-light and baby, the authors claim an optimal action could be found within 1 second.  What kind of problems could be solved as well, if e.g., 10 minutes were allowed?

The submission title refers to "anytime deterministic guarantees", but aside from a single claim that this is achieved, no additional discusssion of the meaning of this term is provided.

**Questions:**

In addition to any possible question that has sneaked in the previous forms, I have additional questions


- in line 156, the first clause of the sentence seems incomplete "In the event that no subsequent chosen observations [...], the value of Q etc";  what condition do the subsequence chosen observations have satisfy?

- in corollary 1.1, it's not very clear what are the conditions for the convergence to work out.  I assume this is the convergence as \bar Z -> Z.  is this right?

- in line 233, it is said that the complexity attributed to the Bellman upate is linear due to the max and expectation operators.  Since the operators are nested and not independent, shouldn't the complexity be quadratic?

# Minor notes and comments about notation

- although clear enough from context and formalization, it might be worthwhile to make a explicit comment about the fact that deterministic policies are implied by the formalization.

- the expectation in equation (1) should be conditioned on b_t.

- the notation in line 89 "\pi_t \equiv \pi_t(b_t)" is a bit confusing;  you're now using \pi_t to refer not just to the policy, but also to the action determined by the policy.  why not use a_t instead, as that's how a_t was already appropriately defined?

- in equation (7), is the simplified expectation suppsed to be of the next simplified belief \bar V(b_{t+1}) ?  If so (as it seems), then the expectation should be conditioned on the current belief.

- in equation (7), is it correct that the expectation  is not conditioned on anything?  it seems like that might be correct just due to a combination of the deterministic actions and how simplified expectation is defined, but a clarification would be helpful.  Arguably, the notation of simplified expectation is confusing, and lines (7-8) would be simpler if an explicit sum is used directly.

- in equation (8), it seems like \bar b(z_{t+1}) is trying to denote the updated belief after integrating the current belief with the new observation z_{t+1}, but this notation was not introduced, and seems in conflict with the notation of b(x) denoting the probabiliy of a state according to the belief.  I think overloading the symbol b this way is fine, as long as both uses have been previously defined appropriately.

- in equation (9), R_max is not defined.  Supposedly this is either the maximal absolute reward achievable by a reachable state-action pair, or the maximal absolute reward achievable by a reachable belief-action pair.  Either way, it should be defined earlier in the preliminaries section.

- in the preliminaries section, the pomdp is defined in terms of R, which supposedly is the reward function, but is never defined.  Later in the section, rewards are instead expressed as r(b, a) for beliefs and r_x(x, a) for states.  In later sections, state rewards are only identified as r(x, a), according to their inputs.  This is fine, but the notation should remain consistent.

- the bound term defined in equation (11) includes the subscript z, in contrast to the one defined in equation (9).  What is the significance of this notation difference?  Similarly, the bound term defined in equation (17) inclues the subscript "x,z".  Including the initial bound denoted with the subscript 0, there are not 4 different notations used, which can be confusing, so a clearer explanation would help.  Anything as simple as explicitly describing that e_0 is the initial bound term, e(b) is the belief-only bound term, e_z(b, a) is the belief-action bound term using simplified observations only, and e_{x,z} is the belief-action bound term using simplified states and observations.

- in many equations, the notation for expectation (and limited expectations) appears to be inconsistent, as sometimes the expectation conditions on an action and sometimes it doesn't.  It appears like this may simply be the case the conditional action is shown explicitly when it is not the one which would be chosen deterministically by the policy \pi, while it is implied when it is the action chosen by the deterministic policy \pi, but it is not clear whether this is really the case.  I urge the authors to standardize their notation as to avoid confusion and maximize comprehension for the reader.

- Algorithm (1) is never referred to in the text, or described.  Further, both algorithm (1) and (2) have the same name `Algorithm-A`.  Please describe the contents of both.

**Limitations:**

As noted by the authors themselves, their proposed method had additional requirements compared to other similar methods, specifically the ability to not just simulate/sample states and observations, but also to compute the respective probabilities.  Although this does not automatically disqualify the proposed method, its validity, and the authors' contributions, it is a significant requirement.

---

> ### Author Rebuttal · Authors · 2023-08-08
>
> - **Limited state and observation sets** - In the theoretical analysis section, we provide flexibility in determining both the limited state and observation sets, enabling us to generalize the results for various algorithms, as discussed in the methods section. Indeed, the reviewer accurately points out a missing clarification concerning the case of a belief in a posterior node, where all states have zero-density value. However, such a situation does not necessitate a restrictive assumption on the set. This is because, in the case the reviewer suggests both the nominator and the denominator of equation (6) nullify. A statement in the spirit of the ones in rows (128-130) will be integrated to the final manuscript regarding the state space as well.
>
> - **Error bounds** - The bounds are problem- and algorithm-dependent and may be loose or tight. For instance, in the experimental section, specifically in Figure 2, the bounds are initially loose but gradually tightened to the optimal value over time. Generally speaking, deterministic bounds will always be at least as large as the original values before performing any calculation. Still, an adaptive algorithm, such as the one used to generate the data for Figure 2, can recover the optimal policy with an increasing number of iterations.
> Clearly, when recovering the entire reachable belief space, the optimal value is obtained. One can construct a POMDP problem where any approximate solver with deterministic bounds only guarantees optimality with the complete state and observation sets; consider an extreme scenario where all actions have equal value. However, this is generally not required with the use of UDB. Whenever the deterministic upper and lower bounds of two candidate actions do not overlap, there is a clear distinction which action is suboptimal, and its subtree needs no further exploration, as implied in the experimental section, lines (297-299); This is in contrast to current state-of-the-art online POMDP algorithms, where any finite-time stopping condition will not guarantee optimality, as the bounds in those algorithms are probabilistic. We agree with the reviewer that this could be made more explicit for better presentation.
>
> - **Evaluation** - The toy Tiger POMDP problem is required by the need to demonstrate our algorithm convergence to the theoretical value obtained via an exhaustive search algorithm, which is highly inefficient in terms of computational burden and will take much longer to converge to the optimal value. Note that the green-dashed line in Figure 2 corresponds to the optimal value obtained by the exhaustive search, as described in the figure caption and lines (269-273).
> As the allowable planning time increases, more challenging POMDPs can be tackled. The time it takes to find the optimal immediate action or policy is problem-dependent.
>
> - **Anytime deterministic guarantees** - This title reflects the contribution of the paper, which are deterministic guarantees of an obtained policy with respect to the optimal value function, which can be obtained anytime, as shown in algorithm-A. Further discussion will be added to the revised version.
>
> - **Q1 - line 156** - No subsequent chosen observation refers to the case where the subtree of a given posterior node is not explored. Thus, the following observations don't have to satisfy any conditions apart from not being chosen for further exploration.
>
> - **Q2 - corollary 1.1** - Although this may happen, in practice, this is not required but only in worst-case scenarios, where the lower and upper bounds continue to overlap in all belief-action nodes until the theoretical tree is fully recovered. As discussed in the comment regarding the \textit{"error bounds"}, often there will be no overlap between bounds for two candidate actions, thus, further exploration in a suboptimal action is not required.
>
> - **Q3 - line 233** - The linear complexity discussed in line 233 only referred to Monte-Carlo solvers, such as POMCP. It is shown in the original paper by Silver et al. [1] (Algorithm 1, procedure Simulate()), that visitation to a posterior node will induce at most $O(|A|)$ complexity, where $|A|$ is the cardinality of the action space. Applying our bounds to POMCP will hinder an additional term to the linear complexity due to the summation over the next observations, as shown in algorithm 2 (lines 19-23). This will result in a linear complexity as well, $O(|A|+|\bar{\mathcal{Z}}|)$, where $|\bar{\mathcal{Z}}|$ is the simplified observation set. Note that our bounds do not require nested summation at any point of the Simulate procedure, as it updates the bounds incrementally. Similarly, POMCP does not require a complete belief update, which would otherwise require nested summation.
>
> - **Minors** - Minors and notation comments will be considered in the final submission.
>
> [1] David Silver and Joel Veness. Monte-carlo planning in large pomdps. In Advances in Neural Information Processing Systems (NIPS), pages 2164–2172, 2010.

---

> > ### Comment · Reviewer_JfUf · 2023-08-21
> >
> > I thank the authors for their clarifications;  Having read the rebuttal, to mine and other reviews has addressed some of the concerns and questions that I had.
> >
> > I'm still not fully convinced I understand the resolution proposed to the issue of limited state and observation sets;  I assume in practice the resolution simply becomes that the relevant probability values remain 0, and if the state and observation sets are incompatible in the way described in the review, then the limited values will also become similarly limited to the immediate reward only, without any subsequent values possible.  A clarification on this subject may be useful to help the reader fully understand how the mathematical formalism does not necessarily require any restrictions on limited state and observation sets, e.g., possibly an example in the appendix, if concerns over limited space exist.
> >
> > My general impression of the technical aspects of the paper has improved a bit, but I'm still quite concerned that there are many many presentation and formalization issues that need to be fixed.  I will improve my recommendation to the chairs, but strongly invite the authors to not overlook these issues, and do a thorough reworking of the manuscript, especially as other reviewers have also pointed out similar issues.

---

### Official Review · Reviewer_9qkM · 2023-07-28

**Soundness:** 3 good
**Presentation:** 3 good
**Contribution:** 3 good
**Rating:** 6
**Confidence:** 2

**Summary:**

The paper introduces novel contributions in the form of deterministic bounds for discrete state space POMDPs and an algorithm to effectively implement these bounds in existing algorithms. The authors' key innovation lies in the identification of a specific subset within the observation and state space.

Through experimentation, the results demonstrate that incorporating these bounds leads to notable performance improvements in the majority of cases.






**Strengths:**

The paper's presentation is commendable, offering clarity, and the derivations appear to be accurate. The inclusion of pseudo-code facilitates easy comprehension of the algorithm.

The presence of theoretical guarantees adds value to the research, instilling confidence in the proposed approach.

**Weaknesses:**

The authors' assumptions in this paper include a discrete state space and a known transition and observation model, which may not always align with real-world practicality.

Certain notations concerning reward functions are found to be inadequately defined, requiring further clarification.

Regarding the experimental results, the substantial gap observed on the Discrete Light Dark scenario is mentioned but lacks in-depth discussion, leaving room for further elaboration.

**Questions:**

See weaknesses

**Limitations:**

The limitation is addressed.

---

### Author Rebuttal · Authors · 2023-08-08

We thank all reviewers for their constructive review, which will help us improve the paper's presentation. Additionally, we acknowledge some missing notations and clarifications, which will be fully addressed in the final manuscript.

---

### Decision · Program_Chairs · 2023-09-21

**Decision:**

Accept (poster)

**Comment:**

This paper derives and integrates deterministic bounds in online POMDP planning algorithms (such as POMCP and DESPOT).

As pointed out by the authors, current guarantees are only probabilistic so developing stronger bounds for online planning problems is an important problem. As a result, the method is theoretically interesting and the resulting integration into DESPOT creates a practical algorithm with strong performance.

Nevertheless, there are concerns about the presentation and experiments in the paper. As discussed by the reviewers, the paper should improve the clarity of the approach as well as the assumptions needed for the method. Results on larger problems would greatly strengthen the paper but, at the least, the current results should be clarified with improved discussion.

The author response was helpful about these issues but there are significant updates that need to be integrated into the paper.